# Longitudinal Follow-Up of Gross Motor Function in Children with Congenital Zika Virus Syndrome from a Cohort in Rio de Janeiro, Brazil

**DOI:** 10.3390/v14061173

**Published:** 2022-05-28

**Authors:** Tatiana Hamanaka, Carla Trevisan M. Ribeiro, Sheila Pone, Saint Clair Gomes, Karin Nielsen-Saines, Elizabeth B. Brickley, Maria Elisabeth Moreira, Marcos Pone

**Affiliations:** 1National Institute of Women, Children and Adolescents Health Fernandes Figueira, Oswaldo Cruz Foundation (IFF-Fiocruz), Rio de Janeiro 22250-020, Brazil; tatiana.hamanaka@iff.fiocruz.br (T.H.); sheila.pone@iff.fiocruz.br (S.P.); saintclair.junior@iff.fiocruz.br (S.C.G.); bebeth@iff.fiocruz.br (M.E.M.); marcos.pone@iff.fiocruz.br (M.P.); 2David Geffen School of Medicine, University of California, 10833 Le Conte Ave, Los Angeles, CA 90095, USA; knielsen@mednet.ucla.edu; 3London School of Hygiene & Tropical Medicine, Keppel St., London WC1E 7HT, UK; elizabeth.brickley@lshtm.ac.uk

**Keywords:** *zika virus*, congenital zika virus syndrome, motor development

## Abstract

Knowledge of how congenital Zika syndrome (CZS) impacts motor development of children longitudinally is important to guide management. The objective of the present study was to describe the evolution of gross motor function in children with CZS in a Rio de Janeiro hospital. In children with CZS without arthrogryposis or other congenital osteoarticular malformations who were followed in a prospective cohort study, motor performance was evaluated at two timepoints using the Gross Motor Function Classification System (GMFCS) and the Gross Motor Function Measurement test (GMFM-88). Among 74 children, at the baseline evaluation, the median age was 13 (8–24) months, and on follow-up, 28 (24–48) months. According to GMFCS at the second timepoint, 6 children were classified as mild, 11 as moderate, and 57 as severe. In the GMFM-88 assessment, children in the severe group had a median score of 10.05 in the baseline evaluation and a follow-up score of 12.40, the moderate group had median scores of 25.60 and 29.60, and the mild group had median scores of 82.60 and 91.00, respectively. Although a small developmental improvement was observed, the motor impairment of children was mainly consistent with severe cerebral palsy. Baseline motor function assessments were predictive of prognosis.

## 1. Introduction

The state of Rio de Janeiro, Brazil, was severely affected by the emergence of Zika virus (ZIKV) in 2015. Although the epidemic waned, ZIKV continues to circulate at low levels. In the first six months of 2019, the Rio de Janeiro State Health Department reported 1387 probable ZIKV cases, with 870 confirmed clinically or by laboratory diagnosis. Of the confirmed cases, 21% (*n* = 180) represent notifications in pregnant women [1].

ZIKV is intensely neurotropic. Congenital infections with ZIKV can interrupt cerebral embryogenesis and result not only in microcephaly (MC), but also in a range of neurological and neurodevelopmental abnormalities. Other common clinical features observed in children congenitally infected with ZIKV include epilepsy, dysphagia, hearing and visual impairments, and osteoarticular abnormalities. This wide spectrum of structural defects and functional impairments is collectively recognized as Congenital Zika Syndrome (CZS) [2,3,4,5,6,7,8,9,10].

Although the characteristic features of central nervous system (CNS) impairment in children with CZS, especially changes in tone and posture consistent with those of children with cerebral palsy (CP) [11], are increasingly well described, CZS-related functional impairments and their evolution over time remain largely unknown. There are currently no quantitative studies describing the impact of CZS on gross motor performance over time.

As children with CZS grow, it is essential to monitor their motor development using standardized assessment tools [2,12]. The Gross Motor Function Measure (GMFM) was developed to assess longitudinal changes in gross motor function in children with CZS [13]. When used in parallel with the Gross Motor Function Classification System (GMFCS), both measures can inform motor prognosis and guide clinical management [13].

We previously performed a cross-sectional analysis of gross motor function in children with CZS in Rio de Janeiro [13]. Building on this work, the present study describes the evolution of gross motor function in children with CZS who have been prospectively followed in our pediatric infectious diseases service at a referral hospital in Rio de Janeiro, Brazil.

## 2. Materials and Methods

The study was conducted at the National Institute of Women, Children and Adolescents Health, Instituto Fernandes Figueira, Oswaldo Cruz Foundation (IFF-Fiocruz), a major referral center for congenital infections in Rio de Janeiro, Brazil. Infants with confirmed exposure to ZIKV or a confirmed clinical CZS diagnosis were enrolled in a longitudinal cohort during the ZIKV outbreak in Brazil in 2015–2016 and followed until the present [12].

The study population included children born during or immediately after the ZIKV outbreak in Rio de Janeiro, Brazil, with presumed CZS (i.e., with or without laboratory confirmation of maternal or fetal infection) as determined by the presence of characteristic clinical features, neurological manifestations, neuroimaging abnormalities, and/or ocular alterations typical of congenital ZIKV infection [3]. The following features characteristic of the clinical phenotype of CZS were observed: microcephaly, overlapping cranial sutures, partially collapsed skull, prominent occipital bone, redundant scalp skin, and/or severe neurological impairment, with or without accompanying arthrogryposis; neurological manifestations including irritability, hypertonia and spasticity, hypotonia, hyperreflexia, and/or seizures; neuroimaging alterations including: intracranial calcifications, ventriculomegaly and extra-axial fluid, abnormal gyral pattern (e.g., polymicrogyria), decreased cerebral parenchymal volume, cortical atrophy and malformations, cerebellar or cerebellar vermis hypoplasia, delayed myelination, and hypoplasia of the corpus callosum; typical ocular alterations - hypoplasia or pallor of the optic nerve, increased cup-to-disc ratio, chorioretinal atrophy or scar, pigment mottling, hemorrhagic retinopathy, and abnormal retinal vasculature.

Children with arthrogryposis or other osteoarticular congenital malformations were excluded from the current analysis to reduce confounding factors in motor assessment. Other congenital infections of the TORCH group, such as toxoplasmosis, rubella, cytomegalovirus infection, and herpes simplex virus infection, were excluded by negative serology. Moderate and severe microcephaly were respectively defined as head circumferences at birth of greater than two or three standard deviations below the mean for gestational age and sex (z-scores of ≤−2 or ≤−3). Z-scores at birth were estimated using the INTERGROWTH-21st (http://intergrowth21.ndog.ox.ac.uk; accessed on 4 May 2017) [14,15].

Information on age, sex, head circumference at birth and prenatal ZIKV infection confirmation (i.e., lab confirmation of maternal infection during pregnancy or congenital infection at birth, through real-time reverse transcription polymerase chain reaction (RT-PCR)) was extracted from medical records. Each child was evaluated by a GMFM-trained physical therapist (ICC > 0.98 between trained therapists) and was classified according to the GMFCS-88 at two timepoints. Children with CZS underwent a first evaluation at ≥6 months of age and underwent a second evaluation at ≥24 months of chronological age and at least one year after the first evaluation, during the period between March 2017 and February 2019.

The GMFM-88 tool assessed 88 items in the following dimensions: (A) lying and rolling; (B) sitting; (C) crawling and kneeling; (D) standing; and (E) walking, running, and jumping. Children were also classified into five progressive levels of independence and functionality according to the Gross Motor Function Classification System (GMFCS) total score. Level I indicates the ability to walk without any restrictions. Level II indicates some limitations in gait. Level III indicates children need some assistance to walk. Level IV indicates children need assistive technology equipment to move. Level V is reserved for children with severe movement limitations even with the aid of modern technology and total dependence for performance of routine tasks. For analytical purposes, we classified children at levels I and II as having mild impairment, at level III as having moderate impairment, and at levels IV and V as having severe impairment. GMFM results were described quantitatively according to the standardization of the scale. Results were provided as a percentage within each dimension and as a total percentage [16,17].

Data were stored in an EPIINFO 7 database. The data analysis was descriptive with measures of central tendency and position and measures of dispersion. GMFM results were analyzed by the Gross Motor Ability Estimator software (GMAE). Differences in means and standard deviations for the percentages within each dimension and in total were analyzed by functional impairment group [16,17]. Correlations between the GMFCS functional classification categories and the GMFM-88 total scores (percentage) were evaluated using Spearman’s rank correlation coefficients. All statistical analyses were performed using Statistical Package for Social Sciences (SPSS) version 20.0 with a significance level of α = 0.05.

This study was nested within a prospective cohort approved by the institutional review board of IFF-Fiocruz (Plataforma Brasil, CAAE: 52675616.0.0000.5269). Parents or guardians provided written informed consent for their children’s participation.

## 3. Results

Seventy-four children with CZS were evaluated, of whom 52.7% (39/74) were male, with ages ranging from 8 to 24 months (the median age was 13 months) in the first assessment. In the final assessment, the median age was 28 months (ranging from 24 to 48 months). More than three-quarters of children with CZS (57/74, 77.2%) had severe gross motor impairment, 11 (14.8%) had moderate impairment, and 6 (8%) had mild impairment. There was no significant difference between groups regarding age and sex (Table 1).

Among the evaluated children, 34 (45.9%) had RT-PCR confirmed prenatal ZIKV exposure (i.e., infection in the mother, neonate, or both), and the majority (55.4%) were exposed in the first trimester of pregnancy. Forty children had a clinical diagnosis of CZS without an accompanying laboratory diagnosis. The proportion of children with microcephaly varied across the gross motor function impairment groups (*p* = 0.001, Fisher’s exact test) being most prevalent in the severe group, present in 82.5% (47/57) of cases. Forty (74.1%) of the 54 children with microcephaly had severe MC, and of these, 92.5% (37/40) presented with severe gross motor function impairment (Table 1).

At the time of the first assessment, 77% (57/74) of children were undergoing physical therapy. At the time of the second assessment, 92% (68/74) of children were undergoing physical therapy. Physical therapy was initiated at a mean age of five-months (range: 1 to 16 months).

In the first assessment, the median GMFM score was 82.6 (ranging from 74.9 to 90.3) in the mild impairment group, 25.6 (19.3 to 37.3) in the moderate impairment group and 10.5 (3.7 to 22.0) in the severe impairment group. There was a statistically significant difference between groups (*p*-value < 0.001, ANOVA). Overall, there was an improvement in scores by the time of the second assessment with a median GMFM score of 91.0 (ranging from 79.9 to 94.7) in the mild impairment group, 29.6 (from 25.9 to 68.3) in the moderate impairment group and 12.4 (from 3.5 to 28.7) in the severe impairment group (Table 2). In the second assessment, a statistically significant negative correlation between GMFCS groups and the GMFM score was noted (Spearman’s rho: −0.732; *p*-value < 0.001). The overall median score in the first GMFM assessment was 11.5 (range: 3.7–90.3), while the overall median score in the second assessment was 14.2 (range: 3.5–94.7).

In regards to the magnitude of GMFM findings, children in the severely impaired group (77.2% of the cohort) only scored in dimensions A (lying and rolling; median score = 49.0) and B (sitting; median score = 16.7). This underscores the high degree of incapacity in the severely impaired group, as these children lacked cervical or trunk control. In comparison, children in the moderately impaired group had higher scores in dimensions A (lying and rolling; median score = 92.9), B (sitting; median score = 55.0), and C (crawling and kneeling; median score= 11.9). In the moderately impaired group, children generally managed to remain seated without upper limb support, and some even had additional abilities, such as rolling and crawling. Children in the mildly impaired group had fewer limitations, and a subset were able to reach some abilities within dimension E (walking, running, and jumping; median score = 57.6). Their difficulties were in the performance of more complex GMFM tasks, such as “to jump 10 times on the same foot in a circle” (Table 3).

## 4. Discussion

In the subset of children with CZS analyzed in this study, we observed little progression in motor development in the first two years of life. In general, the motor development in this population remained compromised despite the fact that most children were enrolled in physical therapy for at least one year following the initial assessment. Most children in this study had severe motor impairment (i.e., GMFCS level V or IV), indicating difficulty in maintaining set postures, such as sitting, or undergoing postural changes and acquiring gait. This group of children, therefore, needs to depend on wheelchairs and guardian assistance for their locomotion. Only 7% of the children assessed had achieved the ability to walk without assistance.

This level of functionality is similar to previous reports by Melo [18], Ventura [19], Carvalho [20] and Cavalcante [21], which found high proportions of children in the severe GMFCS levels IV and V with only a small proportion of less affected children (GMFCS levels I and II). All three studies were based in cohorts from the northeast region of Brazil, although our results indicate there seems to be no regional differences. One recent study which used GMFCS assessments, demonstrated that among 110 children born with CZS, 90.2% were classified as GMFCS level V, 4.9% as GMFCS level IV, 2% as GMFCS level III, and 2.9% as GMFCS level I [20]. This study, however, included children with arthrogryposis, which may have led to a higher prevalence of severe impairment. We did not include children with arthrogryposis in the current analysis to avoid confounding of the GMFM scoring. The literature has previously described the association between congenital orthopedic abnormalities and worse neurological outcomes in CZS [17,20].

Regarding additional parameters, such as sex, age, and microcephaly, findings were similar to those of previous reports [2,22,23,24,25]. We found no differences in motor performance in children by sex or age. The high degree of concordance between severe microcephaly and severe gross motor function impairment has been previously described in the literature [19,20] and likely reflects neurological involvement [2,22,23,24,25]. Microcephaly has been shown to be a marker of worse motor and cognitive functions, with a statistically significant association between head circumference and the presence of cerebral cortical lesions [18,20].

More than half of our cohort did not have RT-PCR confirmation of ZIKV infection Therefore, it was not possible to identify the period of exposure to maternal infection particularly among those whose mothers had asymptomatic infection. Of those with known timing of maternal infection, the majority (55.4%, 41/74) were exposed in the first trimester of pregnancy. First trimester of pregnancy exposure was particularly prevalent in children in the severe group. A prior study of children from our cohort [26] demonstrated that the odds of abnormal neuroimaging findings were nearly eight times greater in children exposed to ZIKV in the first trimester of pregnancy as compared to later trimesters. Exposure to ZIKV during pregnancy has been shown to be similar to other congenital infections, such as cytomegalovirus and rubella, as there is an association between first semester exposure and more severe clinical manifestations due to the viral teratogenic potential during the period of fetal development [27].

Overall, most children in this study had a low GMFM score, indicating functionally significant impairment. Interestingly, the median initial GMFM scores in our children between 8 and 24 months of age were higher than that described by other researchers who evaluated children in a similar age range (e.g., 11.5 in our series versus 6.5 in a prior study by Melo, et al. [18]). In our second assessment, we observed a slight evolution of the gross motor function, with a median score of 14.2 (range: 3.5–94.7). This small scale of improvement in GMFM scores over time was probably due to the large number of children in the severe category who had low GMFM scores at baseline. An improvement was also noted over time in the study by Ventura et al. [19].

The increase in median GMFM scores can be explained by the children’s young ages at the time of performance of the assessments and neuroplasticity, which is greater in the first three years of life. Scores may also be improved over time through motor and sensory stimulation that may contribute to modification and reorganization of functional cortical areas [28]. Children with neurological dysfunction may show some degree of developmental improvement in the first years of life, especially if they are included in early stimulation programs [29,30].

Implementation of physical therapy in over 90% of cases was a positive finding in our study. This is in line with recommendations of the Brazilian Ministry of Health and state guidelines. The State of Rio de Janeiro developed, through their Health Department (SES/RJ), a management plan for CZS cases in the state of Rio de Janeiro [31]. In this plan, the main objectives were to confirm or rule out CZS among suspected cases, and to refer children and their families to multidisciplinary care within the Single Unified Health System (SUS) in Brazil. After this plan was implemented at the state level, an increase in the number of registered cases of children with CZS in the state assistance network was observed [1].

The assessment of motor development is important for understanding neuro-sensorimotor prognosis. Rosenbaum in 2002 identified motor prognostic curves for children with cerebral palsy (CP), in which it is possible to assess gross motor function in relation to mean age and GMFCS level. A wide margin of variability was demonstrated within the expected limits of gross motor function at each level [32].

Since the majority of children in this study were classified as being in the severe group, the assessments indicated a very limited prognosis, with the need for permanent help from a caregiver and wheelchair for mobility [32]. The scores in dimensions A and B of the GMFM after two years of age showed that these children had difficulties overcoming the force of gravity to maintain adequate head control, or that they achieved head control without adequate trunk control. In the group of moderately impaired children, there was a higher score in dimension B and some in dimension C, which demonstrated that they were able to make some postural changes, such as rolling over and even moving to a sitting position and remaining seated with or without support. For this group, based on data generated by Rosenbaum [32] one would expect that regarding the ability to walk (gait prognosis), children could potentially develop this ability with adapted walkers and/or gain mobility using an adapted wheelchair. Children in the mild group scored in dimension E, indicating a good motor prognosis. Some children already performed free walking without the aid of assistive technology, which facilitates their inclusion in the community, including school adaptation.

Our study is one of the largest studies to longitudinally evaluate motor function in children with CZS. Studies with a smaller sample size suggested severe motor impairment in children with CZS. In another study of 34 children with CZS, with a mean age of 21 months, 70% of the sample did not walk and 65% did not have trunk control, while more than 90% showed changes in tone and voluntary movement [11].A separate study of 39 children with CZS reported that motor development at the age of 6 months was equivalent to that expected at 2 to 3 months of age, motor development at 12 months was equivalent to that expected at 3 to 4 months, and motor development at 18 months was equivalent to that expected at 4 to 5 months of age [33].

The negative correlations between GMFCS groups and GMFM-88 scores in follow-up assessments demonstrate that the higher the GMFCS level (greater functional severity), the lower the GMFM-88 score [16,32]. Awareness of this association is important, as determining the level of functionality has important prognostic implications. In addition, awareness that the motor developmental prognosis of children with CZS may be severely limited enables adequate and much necessary planning to manage the health care needs of this vulnerable population. Nevertheless, the current study population had an over-representation of children with microcephaly as compared to the general population of children with CZS; this selection bias could result in an over-estimate of the frequency of severe impairment in children with CZS.

The total number of children followed in our study was generally higher than that of other investigations of motor function in CZS cohorts from other regions of Brazil. Our findings in Rio de Janeiro state were very similar to those described in the northeastern region of the country, an area severely affected by the Zika epidemic. We did not include children with arthrogryposis in the group as this was a disability that would further compromise motor function. This would have further increased the number of participants in the severe motor development group.

## 5. Conclusions

Longitudinal follow-up of a cohort of children with CZS in the state of Rio de Janeiro demonstrated that the delay in motor development of children with CZS is very worrisome. We observed a small degree of improvement in the quantitative motor performance of all severity groups analyzed at one year of follow-up. However, this measurable change was not very clinically important as only 5 of the 74 children were able to walk after 24 months of age. Most children demonstrated poor motor development, which correlates with limited functional prognoses. This highlights not only the need for longitudinal monitoring of motor development through systematic assessments, but also the importance of building a relationship between health, education, and social service actions for implementation and management of comprehensive health care for this vulnerable population of children.

## Figures and Tables

**Table 1 viruses-14-01173-t001:** Description of the study population according to GMFCS classification.

		GMFCS			Total
		MILD	MODERATE	SEVERE	
**Number of participants**		*n* = 6	*n* = 11	*n* = 57	*n* = 74
**Mean (SD) age in months, first assessment**		15.0 (0.7)	13.0 (1.8)	14.3 (4.8)	14.3 (4.5)
**Mean (SD) age in months, second** **assessment**		29.5 (2.8)	27.5 (4.3)	30.2 (5.8)	29.7 (5.4)
**Sex**	Female	2 (33.3%)	6 (54.5%)	27 (47.3%)	35 (47.3%)
	Male	4 (66.7%)	5 (45.5%)	30 (52.7%)	39 (52.7%)
**Microcephaly**	Yes	2 (33.3%)	5 (45.5%)	47 (82.5%)	54 (72.9%)
	No	4 (66.7%)	6 (54.5%)	10 (17.5%)	20 (27.1%)
**Microcephaly severity**	Moderate	1 (50.0%)	3 (60%)	10 (21.3%)	14 (25.9%)
	Severe	1 (50.0%)	2 (40%)	37 (78.7%)	40 (74.1%)
**ZIKV detection** **(RT-PCR)**	Yes	3 (50.0%)	6 (54.5%)	25 (43.8%)	34 (45.9%)
	No	3 (50.0%)	5 (45.5%)	32 (56.2%)	40 (54.1%)
**Trimester of exposure to ZIKV in pregnancy**	1st	2 (33.3%)	9 (81.8%)	30 (52.6%)	41 (55.4%)
	2nd	1 (16.6%)	-	4 (7.0%)	5 (6.8%)
	3rd	1 (16.6%)	-	1 (1.7%)	2 (2.7%)
	Unknown	2 (33.3%)	2 (18.2%)	22 (38.6%)	26 (35.1%)
**Physical Therapy**	Yes	5 (83.3%)	10 (90.9%)	53 (92.9%)	68 (91.9%)
	No	1 (16.7%)	1 (9.1%)	4 (7.1%)	6 (8.1%)
**GMFCS levels**		I = 5 II = 1	III = 11	IV = 47 V = 10	NA

Abbreviations: GMFCS (Gross Motor Function Classification System); RT-PCR (reverse transcription-polymerase chain reaction); ZIKV (Zika virus); SD (standard deviation); NA (not applicable).

**Table 2 viruses-14-01173-t002:** GMFM scores in the first and second assessments by functional level.

GMFCS	GMFM		GMFM	
	First Assessment		Second Assessment	
	Min–Max	Median	Min–Max	Median
Mild	74.90–90.30	82.60	79.90–94.70	91.00
Moderate	19.30–37.30	25.60	25.90–68.31	29.60
Severe	3.70–22.00	10.05	3.50–28.70	12.40

Abbreviations: GMFCS (Gross Motor Function Classification System); GMFM (gross motor function measure).

**Table 3 viruses-14-01173-t003:** Distribution of the sample in relation to the total score of each GMFM dimension of the second assessment and the functional level according to the GMFCS.

	GMFCS Mild (*n* = 6)	GMFCS Moderate (*n* = 11)	GMFCS Severe (*n* = 57)
	Median (Min–Max)	Median (Min–Max)	Median (Min–Max)
Dimension A	100 (100−100)	92.2 (82.4−100)	49.0 (15.7−96.1)
Dimension B	100 (100−100)	55.0 (43.3−100)	16.7 (0−45)
Dimension C	100 (95.2−100)	11.9 (0−95)	0 (0−7)
Dimension D	94.9 (82.1−100)	0 (0−51.3)	0 (0−0)
Dimension E	57.6 (22.2−73.6)	0 (0−13.9)	0 (0−0)
Total score	91.0 (79.9−94.7)	29.6 (25.9−68.3)	12.4 (3.5−28.7)

Abbreviations: GMFCS (gross motor function classification system).

## Data Availability

Data for this study is available upon request.

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
