# Peer review of "Longitudinal Follow-Up of Gross Motor Function in Children with Congenital Zika Virus Syndrome from a Cohort in Rio de Janeiro, Brazil"

_viruses, 2022, doi:10.3390/v14061173_

Round 1
Reviewer 1 Report
This manuscript describes a longitudinal assessment of gross motor function in children with Congenital Zika Syndrome (CZS) as part of a larger study in Brazil. In general, the methods and cohort were well-described and the results were presented clearly. Ongoing evaluation of the development of children exposed to Zika virus are important to understand the long-term consequences of this exposure and the prognosis for children based on early assessments. A few minor issues and suggestions are outlined below:
Given that one of the objectives is to determine the impact of early intervention with physical therapy on motor function development, it would be helpful to offer more analysis of whether there was an interaction between the timing of the initiation of physical therapy and any subsequent improvement in scores on the GMFM or GMFCS assessments. In terms of clinical significance, it could provide some data to evaluate whether there is a correlation between age of beginning physical therapy and degree of improvement within any of the groups (mild, moderate, or severe).
It would be good to spell out more clearly the subset of children with known exposure in the first trimester of pregnancy. In the discussion, it’s mentioned that this exposure time was particularly prevalent in the “severe” group, but the breakdown of this group by score on either/both GMFCS and GMFM would be helpful – perhaps this could be added to Table 1.
It’s a bit surprising that there was a negative correlation between GMFCS groups and the GMFM score only on the second assessment.
Line 134 – It’s not clear where the number of “54” came from without studying the table. I think it is 54 children who were diagnosed with microcephaly, 40 of whom had severe microcephaly. It’s a bit confusing as written and could be revised for clarity.
Minor typos/grammar:
The manuscript should be proofread carefully for grammatical errors and clarity – e.g. line 12 “children’s”. line 13 – “objective”, the sentences in lines 70-80 are not complete sentences, line 139 “in”, lines 263-264 – sentence is not clear.
Author Response
Dear Reviewer,
Thank you for the careful review of our manuscript.
In response to the first comment made by the reviewer, we would like to clarify a bit further the purpose of our article. Both in the abstract and at the end of the introduction, our main objective was to: “describe the evolution of gross motor function in children with CZS who have been prospectively followed up in our pediatric infectious diseases service at a referral hospital in Rio de Janeiro, Brazil”. Although we included data regarding physical therapy management in the manuscript, it was not our intention to determine the impact of early intervention with physical therapy on motor function development in the present analysis. The information about physical therapy was complementary information to inform the reader how the study population was managed, with the purpose of raising additional hypotheses and questions to be addressed in future research and analyses. Describing how participants responded to physical therapy is beyond the current scope of our manuscript. However, we appreciate the reviewer’s suggestions and will incorporate this in our future work.
As for the exposure period – that has been added to table 1
.As for the negative correlation between the GMFCS groups and the GMFM score, the data was sparse and was only analyzed in the second exposure. In the cross-sectional survey we conducted which was mentioned in the introduction, performed in the same patient population, we had described this association and for this reason we did not present it again in the current analysis.
Line 134 (now 138) - The text has been modified to make it more understandable.
Grammatical errors - Our text was reviewed and modified as requested.
Reviewer 2 Report
In the presented manuscript, the authors analyzed the gross motor function in a cohort of 74 children with congenital Zika virus syndrome in Brazil. Please see some comments below.
Abstract
Line 13: please correct "objective".
Introduction
I suggest to add more data on ZIKV, including transmission and clinical presentation.
Materials and Methods
The authors stated that TORCH group infections were excluded by negative serology (lines 83-84). Please add laboratory methods used for ZIKV detection.
Results
Line 129: 34 (45.9%) of children had RT-PCR confirmed prenatal ZIKV exposure. Please clarify and add data for the remining 40 children. How was ZIKV infection confirmed in RT-PCR negative children? Was serology used for ZIKV detection?
Discussion
Line 204: please correct RT-PCR
Lines 206-207: what is 55.4%, X/Y?
References
Please check and correct according to the proposition of the journal.
Author Response
Dear Reviewer,
Thank you for your careful review of the manuscript.
Introduction: Since there are already many articles published on the transmission and clinical presentation of ZIKV in infants, many of which are from our group, we chose to have a more focused introduction related to the present research objective. However we have added more information on the clinical presentation of congenital Zika syndrome to the methods section, which now states (lines 71 – 82):
“The following features characteristic of the clinical phenotype of CZS were observed: microcephaly, overlapping cranial sutures, partially collapsed skull, prominent occipital bone, redundant scalp skin, and/or severe neurological impairment, with or without accompanying arthrogryposis; neurological manifestations - irritability, hypertonia and spasticity, hypotonia, hyperreflexia, and/or seizures. Neuroimaging alterations: intracranial calcifications, ventriculomegaly and extra-axial fluid, abnormal gyral pattern (e.g., polymicrogyria), decreased cerebral parenchymal volume, cortical atrophy and malformations, cerebellar or cerebellar vermis hypoplasia, delayed myelination, and hypoplasia of the corpus callosum; typical ocular alterations - hypoplasia or pallor of the optic nerve, increased cup-to-disc ratio, chorio-retinal atrophy or scar, pigment mottling, hemorrhagic retinopathy, and abnormal retinal vasculature.”
Material and Methods: We have added to lines 93-94 that maternal ZIKV diagnosis or infant diagnosis was based on PCR. We did not use serology for diagnosis. We state now the following: “Information on age, sex, head circumference at birth and prenatal ZIKV infection confirmation (i.e., lab confirmation of maternal infection during pregnancy or congenital infection at birth, through real-time reverse transcription polymerase chain reaction (RT-PCR)) was extracted from medical records.”; This information was only in the results before. Regarding the question on laboratory diagnosis of ZIKV in the infants, in the Results section, we mentioned in the materials and methods that: “The study population included children born during or immediately after the ZIKV outbreak in Rio de Janeiro, Brazil, with presumed CZS (i.e., with or without laboratory confirmation of maternal or fetal infection) as determined by the presence of characteristic clinical features, neurological manifestations, neuroimaging abnormalities, and/or ocular alterations typical of congenital ZIKV infection.” (lines 67 -71). We have included in the results section, to further clarify, a comment that the remainder of the children were diagnosed clinically because of characteristic clinical features of CZS. It is now stated (lines 134-135): “Forty children had a clinical diagnosis of CZS without an accompanying laboratory diagnosis.” The discussion further clarifies that “more than half of our cohort did not have RT-PCR confirmation of ZIKV infection; therefore, it was not possible to identify the period of exposure to maternal infection, particularly among those whose mothers had asymptomatic infection.” (line 208-210).
Discussion: Everything was modified as requested. The fraction in line 206-207 (now 211) referred to (55.4%, 41/74), and this has been corrected in the text.
References: These were edited according to the Journal’s guidelines.